# Leveraging potential of limpid attention transformer with dynamic tokenization for hyperspectral image classification

Dhirendra Prasad Yadav[1,2], Deepak Kumar[2], Anand Singh Jalal[3], Bhisham Sharma📧[4*], Panos Liatsis📧[5*]

1 Department of Computer Engineering & Applications, G.L.A. University, Mathurar, Uttar Pradesh, India, 2 Department of Computer Engineering, NIT Meghalaya, Shillong, Meghalaya, India, 3 School of Computer Science & Information Technology, Devi Ahilya Vishwavidyalaya (DAVV), Indore, Northern Mariana Island, India, 4 Centre for Research Impact & Outcome, Chitkara University Institute of Engineering and Technology, Chitkara University, Rajpura, Punjab, India, 5 Center for Cyberphysical Systems, Department of Computer Science, Khalifa University, United Arab Emirates

* Panos.liatsis@ku.ac.ae (PL); bhisham.pec@gmail.com (BS)

## Abstract

Hyperspectral data consists of continuous narrow spectral bands. Due to this, it has less spatial and high spectral information. Convolutional neural networks (CNNs) emerge as a highly contextual information model for remote sensing applications. Unfortunately, CNNs have constraints in their underlying network architecture in regards to the global correlation of spatial and spectral features, making them less reliable for mining and representing the sequential properties of spectral signatures. In this article, limpid size attention network (LSANet) is proposed, which contains 3D and 2D convolution blocks for enhancement of spatial-spectral features of the hyperspectral image (HSI). In addition, limpid attention block (LAB) is designed to provide a global correlation of the spectral and spatial features through LS attention. Furthermore, the computational costs of LS-attention are less compared to the multi-head self-attention (MHSA) of the classical vision transformer (ViT). In the ViT encoder a conditional position encoding (CPE) module is utilized that dynamically generates tokens from the feature maps to capture a richer contextual representation. The LSANet obtained overall accuracy (OA) of 98.78%, 98.67%, 97.52% and 89.45%, respectively, on the Indian Pines (IP), Pavia University (PU), Salina Valley (SV) and Botswana datasets. Our model's quantitative and qualitative results are considerably better than the classical CNN and transformer-based methods.

## 1. Introduction

Hyperspectral images are acquired through spectrometer sensors that capture several narrow overlapping spectral bands [1]. In an HSI, each pixel is represented by a vector equal to the number of spectral bands. Since every vector component is

**Data availability statement:** Dataset used in the study can be downloaded from https://figshare.com/articles/dataset/Hyperspectral_dataset/29371889.

**Funding:** The author(s) received no specific funding for this work.

**Competing interests:** The authors have declared that no competing interests exist.

measured by matching to a specific wavelength, the pixels have enormously detailed spectral signatures. The contiguous acquisition allows the radiance spectrum to be precisely estimated at each pixel in the image [2]. The extensive spectrum information improves surface feature and object discrimination over standard imaging methods [3]. However, these bands have close relationships due to the short spectral distance and contain redundant information. Since hyperspectral cameras are not built for particular applications, certain beneficial bands may not be helpful in others. As a result, collecting application-specific information is critical for maximizing the benefits of hyperspectral images [4].

The classification of HSIs is a non-linear problem [5], and the initial attempts by linear transformation-based statistical techniques like discriminant analytical methods [6,7], principal component analysis methods, and wavelet transforms [8] do not yield satisfactory results for HS data. In contrast, composite [9], probabilistic [10], and generalized kernel [11] methods demonstrated the potential to produce promising results. Nevertheless, these methods focus on only spatial features for HS data classification. The feature extraction strategies aided by some machine learning algorithms take less time, cost, and space complexity. At the same time, classification performance is not optimal. Following the success of these classical methodologies for HSI categorization, researchers applied the most current emerging computer vision-based model, which made the classification procedure easier and closer to excellence.

According to research, advances in artificial intelligence (AI), from the last decade is considered as the most rapidly evolving era in advanced automated technology. Machine learning (ML) is a sophisticated technology that mimics the cognition of the human brain. By holding abstraction, it simply describes a complex system. As a result, it may decrease complications and delve into the insights of large amounts of HSI to uncover promising spatial and spectral features [12]. Recently, deep learning methods provided promising results for HSI classification. However, local kernel features extracted by CNNs lack the global co-relation of the spectral and spatial features. The ViT improved the global co-relation of the features through the attention mechanism. However, classical ViT fails to perform well on the HSI data. In addition, attention mechanism costs are high.

To overcome these challenges, LSANet is developed, which improved the accuracy of HS data classification. The spectral and spatial features are extracted through lightweight 3D-CNN and 2D-CNN and provided attention through a transformer. CPE generates dynamic positional encoding of the features map in the transformer block. The feature map is divided into rows and columns for parallel LS-attention. In the LS-attention, tokens directly interact within the regions and capture more comprehensive contextual information from the HS data. Further, model performance is evaluated on four datasets and achieves better quantitative and visual results.

The significant contributions of the paper are as follows:

[1] A lightweight 3D-CNN and 2D-CNN is designed for the spectral and spatial features. The 3D-CNN captures spectral features, the 2D-CNN explores spatial features, and a global co-relation of spectral and spatial features is provided using a transformer.

[2] The LSANet contains a CPE module that generates dynamic positional encoding using positional encoding generator (PEG), which is translational equivalence and provides complex positional relationships. In addition, zero padding was added in the CPE to retain knowledge of the position and boundary region.

[3] In the LSA, the feature map is divided into rows and columns for parallel attention calculation in the region through the tokens, reducing computation costs and producing more comprehensive contextual information.

[4] The proposed model is evaluated on four standard datasets, and better classification performance is obtained compared to CNN and transformer-based methods.

The rest of the paper is organized as follows.

In Section 2, a detailed description of the classical, CNN and transformer-based methods has been discussed. Section 3 provides a detailed description of the proposed method. Section 4 includes a detailed overview of the datasets, experimental results, and ablation study. Finally, in section 5, we comprehensively conclude the proposed method.

## 2. Related work

Several methods based on machine learning, CNN and ViT, have been developed to classify the land covers available in the hyperspectral data. Chen et al. [13] utilized the classical PCA method for dimensionality reduction. After that, a LBP (local binary pattern) is applied to extract texture features. Furthermore, the grey wolf optimization technique is used to improve features. Finally, the kernel extreme learning machine (KELM) classifies the objects of the hyperspectral image. Camps and Bruzzone et al. [14] applied kernel-based methods to assess the performance of support vector machines (SVMs), regularized radial basis function neural networks (Reg-RBFNN), regularized AdaBoost (Reg-AB), and kernel fisher discriminant (KFD) analysis. They compared Reg-AB and Reg-RBFNN for HSI classification and achieved high accuracy in noisy environments. The edge-preserving filtering method applied by Kang et al. [15] improved the spectral-spatial features. Their method classified HSI by pixel-wise classifier, and the result is presented through multiple probability maps. Finally, the class of each pixel is selected based on maximum probability. Ratle et al. [16] proposed a Laplacian support vector machine (LapSVM) method for HS data classification. The semi-supervised LapSVM method results are compared with those of a supervised SVM.

Deep learning (DL) methods have recently been developed to classify the HSI. Sun et al. [17] proposed a fully convolutional segmentation network (FCSN) method to identify the land cover labels of all pixels in a HSI cube. First, they demonstrate the weak generalization capabilities using CNN-based methods. After that, provide the label of all pixels in the HSI cube for detailed spatial land-cover distributions. Finally, the method used pixel labels to improve the diversity of spatial feature distributions in the HSI and achieve an average accuracy (AA) of 88.31% on the IP dataset. Wang et al. [18] proposed a unified multiscale learning (UML) model for the classification of land covers. They proposed two mechanisms, spatial channel attention and multiscale shuffle block, to enhance spatial and spectral features in the land covers. Bai et al. [19] urged that due to rich spectral information in HSI, it makes similar spectral curve trends, which makes it challenging in land cover classification. To resolve the issue, they proposed a spectral curve-based method to enhance the spectral features and applied a dual attention mechanism to enhance the spatial features. Liu et al. [20] analyzed the spectral and spatial features of the HSI for pixel-level classification of the land covers. They extracted spectral and spatial features from the central pixel through SDPCA (scaled dot-product central attention). Furthermore, a central attention network (CAN) module is designed to classify the land covers in the three datasets. Paoletti et al. [21] used ghost model architecture with CNNs to reduce the computational cost and achieve an efficient classification performance.

Hong et al. [22] applied CNNs and GCNs (graph convolutional networks) for the HSI classification. The GCN network works on non-grid data representation. Furthermore, they designed a new mini-batch GCN (miniGCN) for training large-scale GCNs. Subsequently, they compared the different HSI features using CNN and GCNs. Hang et al. [23] proposed an attention-aided CNN model for HSI classification. The attention mechanism focuses on more discriminative channels

and the spectral attention subnetwork to improve the land cover classification. Using spectral-spatial attention (SPA), their method obtained an accuracy of 89.76% on the Houston 2013 dataset. Cao et al. [24] applied a deep learning approach for HSI classification using a unified framework. They trained a CNN using labelled pixels. Furthermore, they selected the pixel from the labelling pool. Finally, fine-tuned labelled pixels with a new training set are passed to the Markov random field to enforce class label smoothness and enhance the classification performance. Hou et al. [25] used contrastive learning for hyperspectral image classification. Their method uses the information of generous unlabelled samples to help with insufficient label information in hyperspectral data. They design a two-stage model, which enables the model for positive and negative sample judgement. After that, a small amount of samples is used to extract and fine-tune the features of the hyperspectral image.

Sun et al. [26] proposed a multi-structure KELM with an attention fusion strategy (MSAF-KELM) for the accurate fusion of multiple classifiers that effectively classified the land covers. Furthermore, they applied a weighted self-attention fusion strategy (WSAFS), which merges the KELM sub-branch output and self-attention mechanism to achieve efficient fusion results. Their method obtained an accuracy of 95.64% on the SV dataset using spectral-spatial attention MSAF-KELM. Zheng et al. [27] suggested that cropping of the HSI data may result in the spatial information loss of the input image. They proposed two modules based on a rotation-invariant attention network (RIAN) for the HS data classification. In a central spectral attention (CSpeA) module, they avoided the effects of the other categories of land covers and quashed extreme spectral bands. Furthermore, a 1x1 convolution-based rectified spatial attention (RSpaA) module is utilized to avoid the rotation invariant problem and extract spatial and spectral features for land cover classification. Zhang et al. [28] developed a single-source domain expansion network (SDEnet) model to ensure the reliability and effectiveness of domain extension. They use generative contentious learning to train the source domain (SD) and test the target domain (TD). Semantic encoder and morph encoder generate the extended domain (ED). The overall accuracy of SDEnet for Houston 2018 data is 79.96%. Recently, several ViT-based methods have been utilized to classify HS data. In this regard, Ahmad et al. [29] claim average pooling in ViT may result in information loss. To solve the issue, the wavelet-based attention mechanism was utilized to design WaveFormer. The WaveFormer enhanced the interaction of the tokens between different patches, shapes and channel maps, resulting in better classification performance. Sun et al. [30] proposed a spectral-spatial attention network (SSAN) for HSI classification using spectral- and spatial features. First, a simple spectral-spatial network (SSN) extracts spatial-spectral features. After that, spectral and spatial modules consisting of 3-D convolution and activation functions are applied to reduce the influence of identical neighbour pixels. In another research, Sun et al. [31] introduce a spectral-spatial feature tokenization transformer (SSFTT) for spectral-spatial and high-dimension semantic features. First, they utilized 3-D and 2-D convolutional layers to extract spectral and spatial features. Further, a transformer encoder in which Gaussian weighted tokens are generated for feature transformation and representation. After that, the learnable token of the sample label is classified using a softmax layer. Haut et al. [32] introduce visual attention-driven techniques for HSI classification using residual networks (ResNets) for feature extraction. This method evaluates the mask applied for the feature obtained and identifies the different land covers in HS data.

## Problem statement

Hyperspectral images contain rich spatial and spectral features that are utilized in several applications, including precise agriculture, cancer diagnosis, surveillance, etc. For the land cover classification, pixels are labelled based on the spatial and spectral characteristics. In addition, it poses a high-dimension and complex spectral-spatial correlation that must be carefully utilized for classification. Traditional CNNs excel at exploring spectral and spatial features but lack global contextual information, thus often leading to misclassification of the objects at boundaries and edge regions. At the same time, transformer-based methods provide long range dependency to the feature map but require a high volume of data for training the model. In addition, the computation costs of the classical ViT attention mechanism are quadratic in time. This study designed LSANet with three major components: 3DCNN, 2DCNN, and limpid shape attention-based ViT encoder.

The 3D-CNN is designed using the convolution filter number of 64, kernel size of 3x3, followed by batch normalization (BN) and ReLu activation function. A moderate number of filters, such as 64, ensures diverse features from different filters. At the same time, a small kernel size of 3x3 captures fine-grained spatial features from the edge and boundary region. Moreover, batch normalization stabilizes the training, and ReLu activation allows RSANet to learn the complex patterns of the HSI. The features extracted from the 3D-CNN are reshaped and passed to the 2D-CNN module for spatial feature refinement. The 2D-CNN block contains two convolution layers, one with the convolution filter number of 128 and a kernel of size 3x3, followed by BN and ReLu activation function and zero padding. The second convolution layer has 256 filters, kernel size 3x3 and ReLu activation function. The features obtained from the 2D-CNN block are flattened, and tokens are generated for the ViT encoder. The limpid shape attention block captures rich contextual information by splitting it into multiple limpid shape blocks, each of which has equal numbers of rows and columns to capture self-attention. In addition, LSANet is GPU friendly and parallel computation of attention score can be performed, which results in less training time.

## 3. Proposed method

In the proposed study, the limpid size attention network (LSANet) is designed as shown in Fig 1. It has 3D and 2D convolution blocks to leverage the spectral and spatial features from the HSI. In addition, limpid attention block (LAB) is designed to provide a global correlation of the spectral and spatial features through LS attention. Moreover, a conditional position encoding (CPE) module is incorporated that dynamically generates tokens from the feature maps to capture richer contextual representation.

### 3.1. The convolutional module

Let the HSI image. $I \in R^{PxQxD}$, which has height P, width Q and spectral bands D. Principal component analysis (PCA) is applied to reduce the dimension of the HSI to

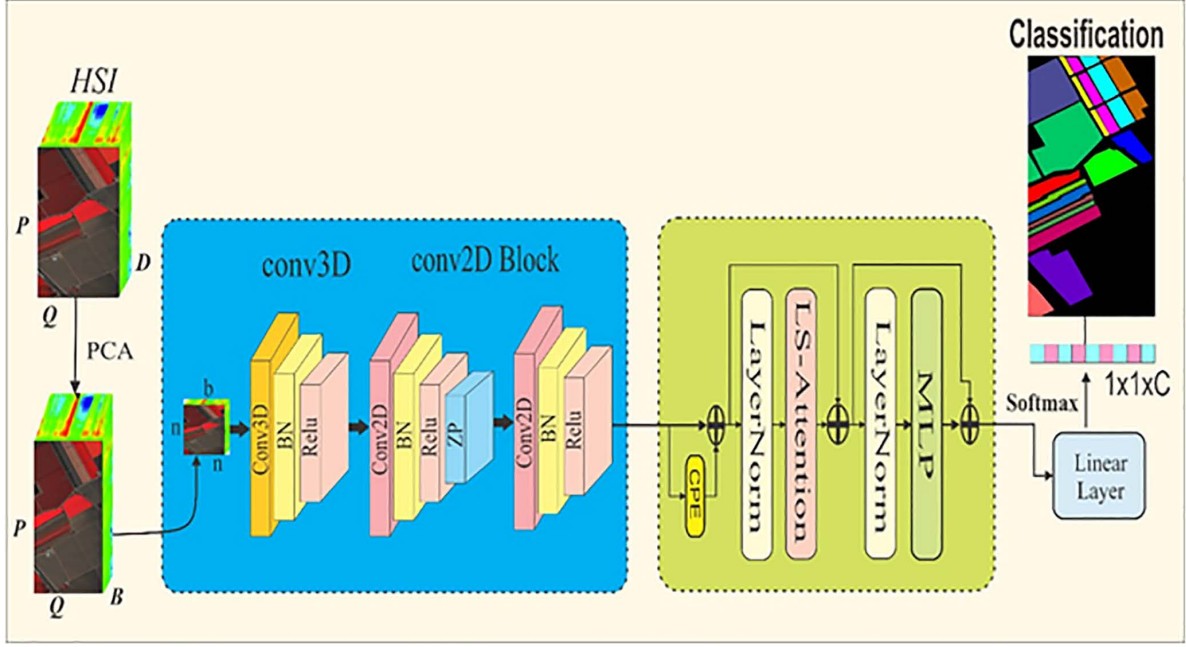

**Fig 1. The proposed LSANet architecture for LULC in HSI.**

$I \in R^{PxQxB}$. After that, patches of dimension $n \times n \times b$ are extracted. Each patch is formed around a pixel, which serves as the patch's focal point. Furthermore, a padding procedure is applied to generate a spatial foundation for these pixels. The HSI contains rich spatial and reduced spectral features. For the efficient segregation of land cover from the HSI, it is necessary to extract spectral and spatial features. The 3D-CNN is capable of extracting spectral features. At the same time, a higher number of 3D-CNN layers increases computational costs. In the proposed study, we utilized a single 3D-CNN layer for spectral and spatial features. The 3D-CNN is designed using 64 convolutional filters, with kernel size of 3x3 followed by BN and ReLu activation functions. A moderate number of filters such as 64 ensures diverse feature extraction from different filters. At the same time, the small kernel size of 3x3 captures fine grained spatial features from the edge and boundary regions. Moreover, batch normalization stabilizes training and ReLu activation allows LSANet to learn the complex patterns of the HSI. The features extracted from the 3D-CNN are reshaped and passed to the 2D-CNN module for spatial feature refinement. The 2D-CNN block contains two convolutional layers, one with 128 convolutional filters, with a kernel size of 3x3, followed by BN and ReLu activation function and zero padding. Zero padding preserves the spatial resolution of the pixel-wise HSI classification. At the same time, it does not increase the computational burden since only zeros are added around the feature map. Moreover, it ensures that the output dimension matches the input. Reflection padding creates mirrors of the edge and boundary regions. However, in HSI, the pixel pattern of land cover differs, and performing reflection padding may lead to overlapping of different class pixels. At the same time, set parameters are used to represent paddings in the learned embedding training. Thus, this type of padding will increase training times and computational burden. Zero padding does not contain meaningful information about the edge and boundary regions, which may reduce the classification performance. ViT encoders have been utilized to mitigate the problem and provide a global correlation to the overlapping patches. The second convolution layer has 256 filters, with kernel size of 3x3 and ReLu activation function. The spatial features extracted in the convolutional layers are calculated as follows.

$$z_{u.v}^{i.j} = f\left(\sum_{m}\sum_{l=0}^{I_i-1}\sum_{r=0}^{J_i-1} w_{l.r}^{i.j.m} * z_{u+l.v+r}^{i-1.m} + b^{i.j}\right)$$

(1)

where $i$ = layer under consideration, $j$ = number of the feature maps in the layer $i$, $z_{u,v}^{i,j}$ = Output features at position (u, v) of the $j^{th}$ feature map in the layer $i$, $b^{i,j}$ = Network bias. The activation function for each layer is denoted by f (.). The m index contains a collection of feature maps from layer ($i$-1) that are the inputs to layer $i$. $w_{l,r}^{i,j,m}$ is the weight in position (l, r) in which the convolutional kernel is related to the $j^{th}$ feature map of the $i^{th}$ layer, $J_i$ and $I_i$ are the kernel's column and row sizes. The feature extraction specifications for the 3-DCNN model is very similar to the 2-DCNN model shown in Eq. (2).

The connection of the spectral dimension is preserved by organizing the related spectral bands in ascending order. The patch extraction and label identification process in the 3D and 2D convolutions are very similar. The 3D CNN block feature extraction procedure is defined as follows.

$$z_{u.v.d}^{i.j} = f\left(\sum_{m}\sum_{l=0}^{I_i-1}\sum_{r=0}^{J_i-1}\sum_{k=0}^{K_i-1} w_{l.r.k}^{i.j.m} * z_{u+l.v+r.d+k}^{(i-1).m} + b^{i.j}\right)$$

(2)

where $K$ is the total number of kernels in layer $i$ and $K_i$ is the size of the 3D kernel in the spectral dimension. Parameter $w_{l,r,k}^{i,j,m}$ is the weight at location (l, r, k), whose convolution kernel corresponds to the $j^{th}$ feature map in the $i^{th}$ layer.

### 3.2. The ViT module

The features obtained from the 2D-CNN block are flattened and reshaped to $Y \in R^{h \times \omega \times c}$. After that, the feature map $Y \in R^{h \times \omega \times c}$ is divided into several limpid blocks $\{L_{1,....} L_N\}$ of the same size $(u_r, u_c)$, where

$L_i \in R^{(u_r, N+u_l, h-u_r, u_r) \times v}, I \in \{1, 2, \ldots, N\}$. The padding technique is used to ensure that the number of pales equal to $N = \frac{h}{u_r} = \frac{w}{u_c}$. In addition, the distances between neighboring rows and columns are identical for all limpid blocks. Furthermore, the self-attention is carried out within each limpid separately. Fig 2 shows that compared to all preceding local self-attention processes, the ILS attention's receptive field is substantially larger and richer, providing a more effective ability for context modelling.

The LS-attention is divided into two parallel row and column paths to reduce the computational costs. Furthermore, row and column wise self-attention mechanism improves token interaction in the groups. The feature map $Y \in R^{h \times w \times c}$ breaks into two halves $Y_r \in R^{h \times w \times \frac{c}{2}}$ and $Y_c \in R^{h \times w \times \frac{c}{2}}$ in the channel dimension and it is defined as follows.

$$Y_r = \left[ Y_r^1, \ldots, Y_r^{N_r} \right], Y_c = \left[ Y_c^1, \ldots, Y_c^{N_c} \right], \tag{3}$$

where $N_r = h/u_r, N_c = w/u_c$. $Y_r^i \in R^{u_r \times w \times c}$ and $Y_c^j \in R^{h \times u_i \times c}$ contain $u_r$ interlaced rows and $u_c$ interlaced columns, respectively. SA is then applied to each group of tokens organized by row and column, respectively. Furthermore, three convolution layers, $\varphi_Q, \varphi_K$, and $\varphi_V$ are used to generate the query, key, and value as follows.

$$Z_r^i = MSA \left( \varphi_Q \left( Y_r^i \right), \varphi_K \left( Y_r^i \right), \varphi_V \left( Y_r^i \right) \right) \tag{3}$$

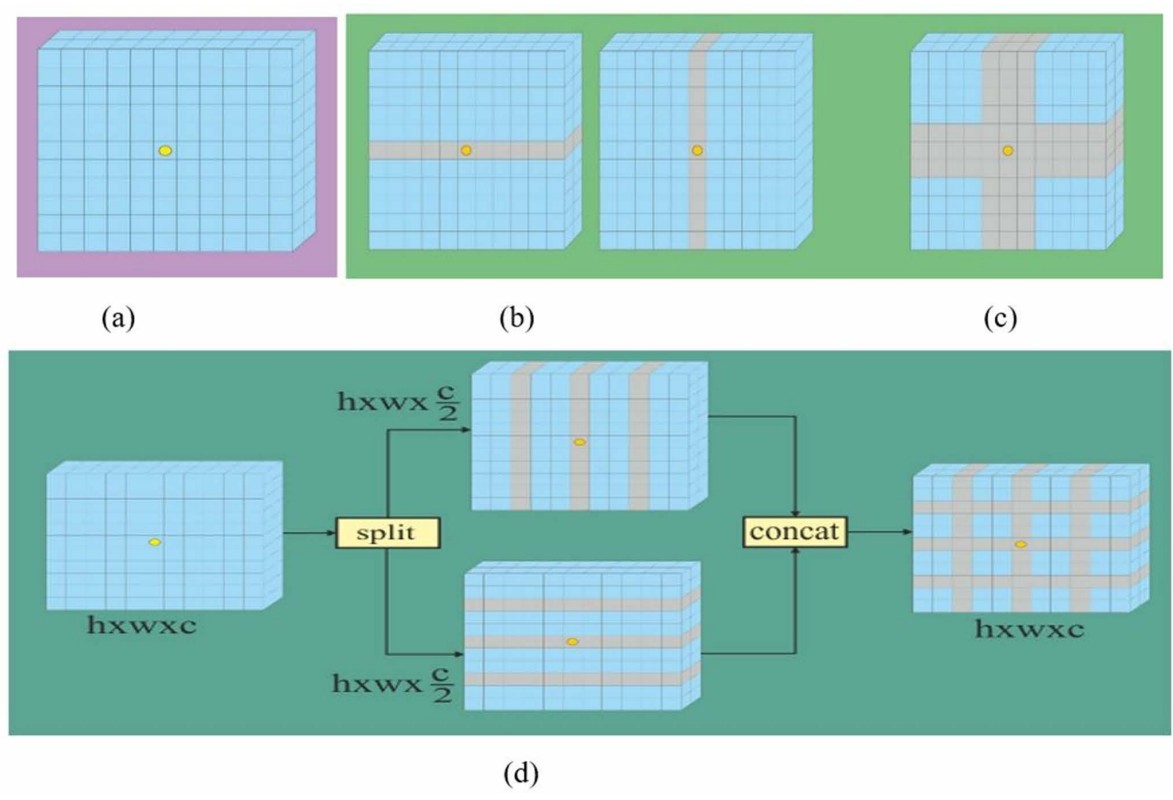

**Fig 2. Illustration of standard GSA (2(a)), axial self-attention (2(b)), cross shaped attention (2(c)) and the proposed LS-attention (2(d)).** In 2(b), 2(c) and 2(d), the shadow area represents the input features split into different groups on which SA is conducted, and the yellow dot can directly interact with the token covered in the shadow region. Here, $h \times w \times c$ represent height, width and channel of the image, respectively.

$$Z_c^i = MSA\left(\varphi_Q\left(Y_c^i\right), \varphi_K\left(Y_c^i\right), \varphi_V\left(Y_c^i\right)\right) \tag{4}$$

The final output $Z \in R^{h \times \omega \times c}$ is generated by concatenating $Y_r^i$ (row-wise) and $Y_c^i$ (column-wise) attention along the channel dimension as follows.

$$Z = Concat\left(Z_r, Z_c\right), \tag{5}$$

where $Z_r = \left[Z_r^1, ..., Z_r^{N_r}\right]$ and $Z_c = \left[Z_c^1, ..., Z_c^{N_c}\right]$.

### 3.2.1. Limpid Attention Block (LAB).

In the ViT, self-attention is calculated on the patches to provide a global correlation of the spatial features. Several studies reported that their ViT model, e.g., Swin transformer, has a square window-based self-attention mechanism. Axial attention-based ViT calculates attention row and column-wise. In the proposed study, the LSANet attention block is named limpid-shaped attention, which is non-square and calculates attention on the tokens. The limpid-shaped attention has broader receptive fields and linear time complexity. Our LAB is composed of the CPE that dynamically generates the positional embedding using PEG. The global attention of the spectral and spatial features is provided using the LSA module that collects contextual information and the MLP that captures complex patterns in the hyperspectral image. The $l^{th}$ block's forward pass can be written as follows:

$$\widetilde{Y^l} = Y^{l-1} + CPE\left(Y^{l-1}\right), \tag{6}$$

$$\hat{Y^l} = \widetilde{Y^l} + LS - Attention\left(LN\left(\widetilde{Y^l}\right)\right), \tag{7}$$

$$Y^l = \hat{Y^l} + MLP\left(LN\left(\hat{Y^l}\right)\right), \tag{8}$$

where LN(.)=layer normalization.

The global self-attention (GSA) model sets a global token to capture the global co-relation of the features (Fig 2(a)). However, it may miss the local information and computational costs also increase. On the other hand, axial self-attention captures the information through local regions via row wise or column wise (Fig 2(b)). It reduces the computation costs and at the same time it limits the model interaction with other parts of the input token sequence. In the cross shaped attention, the model can interact with several rows and columns of the input sequence to improve the features (Fig 2(c)). However, interacting with local neighbours restricts the model's capability in certain applications. In addition, selecting the window size for the specific application is crucial to attaining better performance. The limpid shape attention block captures rich contextual information. The feature map is split into multiple limpid shape blocks in which each limpid has equal rows and columns to capture the self-attention. In addition, it is GPU friendly, providing for parallel computation of attention scores, which results to reducing the training time. The yellow shadow shown in Fig 2(d) represents one limpid that interacts with the other tokens in the same limpid. Furthermore, a parallel LS-attention is implemented for more contextual information.

In traditional ViT, positional encoding is performed by either learned position embedding or fixed positional encoding techniques (Fig 3(a)). However, these techniques produce fixed-length input encoding on which the model is trained.

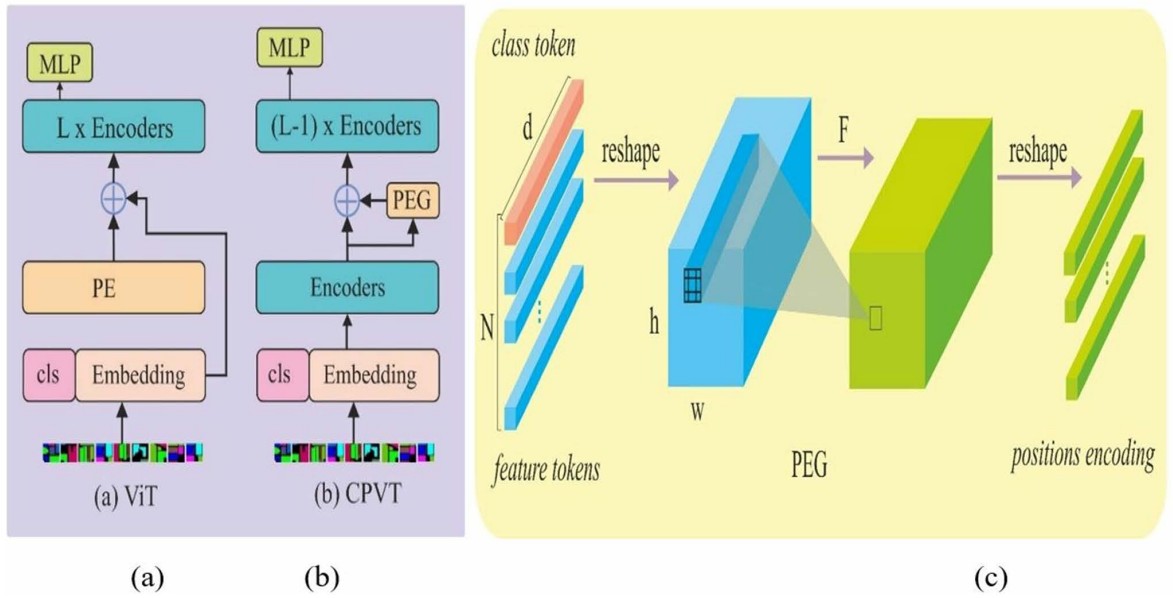

**Fig 3. Illustration of (a) standard embedding in ViT, (b) proposed CPE, (c) PEG block.**

During testing of the model, this causes difficulties for data with longer sequences. In the proposed method, a positional invariant encoding scheme, CPE, was used to generate the input sequence (Fig 3(b)). The CPE utilizes a depth-wise convolution in the PEG shown in Fig 3(c). The flattened input sequence in the PEG $Y \in R^{h \times w \times c}$ is reshaped to 2D image space $Y' \in R^{b \times h \times w \times c}$. After that, a function depth-wise convolution block of a 3x3 filter is applied to produce the CPE. In addition, zero padding is applied to the absolute position. The CPE is defined using Equation (9).

$$Y' = reshape(Y, (b, h, w)) \tag{9}$$

where $Y \in R^{h \times w \times c}$ = Flatten input sequence and $Y' \in R^{b \times h \times w \times c}$ = Reshaped 2D space. After that, a depth-wise convolution is applied to the 2D space to generate the CPE.

$$CPE = DepthwiseConv(Y', W_d) \tag{10}$$

$W_d \in R^{3 \times 3xc}$ = Convolution filter applied to each channel. Furthermore, a zero padding is applied on the generated CPE to maintain the absolute position knowledge as follows.

$$CPE_{padded} = ZeroPadding(CPE, pad = 1) \tag{11}$$

Here, $CPE_{padded} \in R^{(b+2) \times (w+2) \times c}$. The final CPE is generated by trimming the padded CPE to the original dimension.

The LSA is described in Eq. (7) and is constructed by utilizing Eq. (3) to (5). Moreover, the MLP described in Eq. (8) consists of two linear projection layers that expand and shrink the embedding dimension. A softmax layer is added to classify the land covers on the top of the model, and loss is calculated using the Categorical Crossentropy function.

The algorithm for land cover classification using RSANet is described below.

---

**Algorithm 1.** The RSANet for land cover classes classification

---

**Input:** HSI $I \in R^{PxQxD}$
**Output:** The predicted label Y
1: Apply PCA on $I \in R^{PxQxD}$ to reduce the dimension into $I \in R^{PxQxB}$.
2: Extracts patches of dimension $n \times n \times b$.
3: For i= 1 to 200 do
 (a) Feed the patches to the 3D-CNN block
(b) Features extracted from the 3D-CNN are reshaped and passed to the 2D-CNN block.
(c) Features extracted from the 2D-CNN blocks are flattened and reshaped to $Y \in R^{h \times \omega \times c}$.
 (d) partition $Y \in R^{h \times \omega \times c}$ into two independent components $Y_r \in R^{h\alpha\omega \times \frac{c}{2}}$ and $Y_c \in R^{h\alpha\omega \times \frac{c}{2}}$
 (e) Generates the Q, K, and V using Eq. (4).
(f) Calculate the LS-Attention and MLP projection using Eq. (6), Eq. (7) and Eq. (8).
End.

---

## 4. Experimental result and discussion

In this section, quantitative and visual results on PU, IP, SV and Botswana datasets and different parameters affecting the performance are discussed in detail.

### 4.1. Dataset

The Pavia University (PU), Indian Pines (IP), Salinas Valley (SV) and Botswana are the four benchmark datasets on which the model was utilized for the experiments. The PU dataset comprises images of $610 \times 340$ pixels, each containing 115 spectral bands. At the same time, it is categorized into 9 land cover types: asphalt, meadows, gravel, trees, metal sheet, bare soil, bitumen, brick, and shadow. The collection comprises a total of 42,776 samples that have been accurately tagged. The IP dataset is a collection of hyperspectral images used for segmenting an area in Indiana, US. It consists of 220 bands of spectral reflectance. The dataset consists of $145 \times 145$ pixels, each representing a $20\,m \times 20\,m$ region. The dataset comprises 16 land cover classes. The SV dataset was acquired utilizing the 224-band AVIRIS sensor across the geographical area of Salinas Valley, located in California. The image size is 512 x 217, with 16 distinct land cover classes, including vegetables, bare soils, and grape fields. The dataset is extensively utilized for hyperspectral image classification applications. The Botswana dataset comprises hyperspectral images that accurately depict the various land cover categories found in the region of Okavango Delta, located in Botswana. The dataset is suitable for hyperspectral image classification, each image pixel labelled according to its spectral signature. The desciptions of the datasets are depicted in Figs 4–7.

### 4.2 Experimental setup

The RSANet is evaluated in terms of performance on the PU, IP, SV and Botswana datasets using a NVIDIA QUADRO RTX-4000 GPU. Python was used for script writing on Windows 10 operating system. For each experiment, the model was trained for 200 epochs using a batch size of 64. Furthermore, the Adam optimizer with initial learning rate of $3e^{-4}$ was used to accelerate the training process.

### 4.3. Quantitative results comparison

The proposed LSANet performance is compared with 2D-CNN [32], 3D-CNN [33], HybridSN [34], CSIL [35], Spectral-Former(SF) [36], DSGSF [37], morphFormer (MF) [38] and SS1DSwin [39], shown in Tables 1–4.The 2D-CNN and 3D-CNN utilized 2D and 3D convolution layers to extract spatial and spectral features. Meanwhile, HybridSN implemented joint 3D-CNN and 2D-CNN to improve land cover classification. Centre-to-surrounding interactive learning (CSIL) utilized

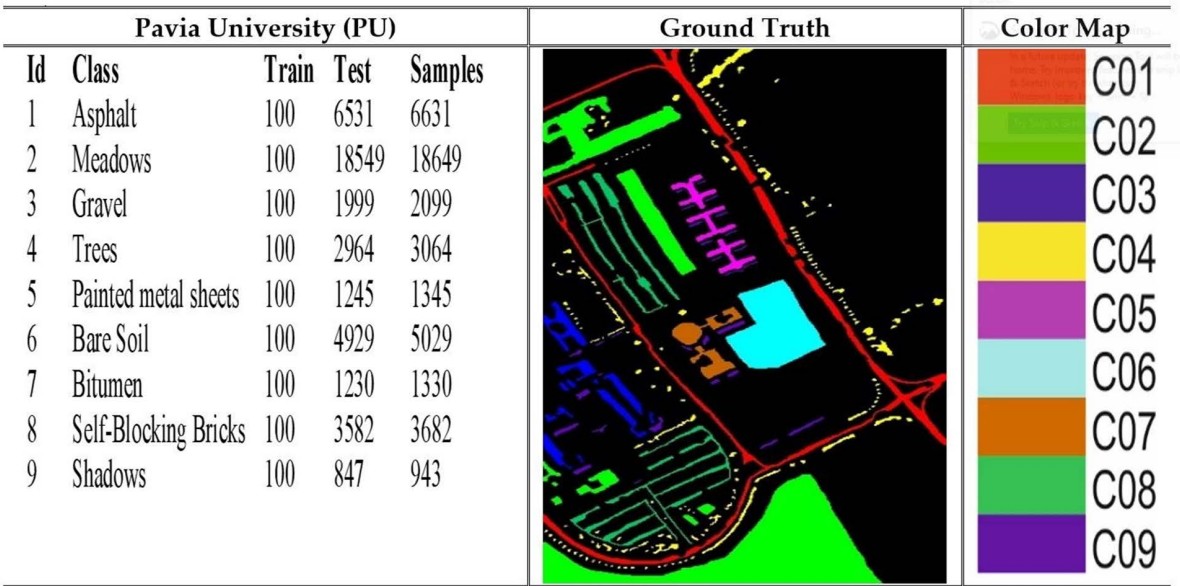

**Fig 4. Details of the Pavia University (PU) dataset with samples and color coding.**

two transformer modules, the first one for central region pixel extraction and avoiding the blur effect in the image. The second surrounding transformer block performed local attention and improved the spatial features globally. In the SF model, HSI classification can be done using pixel-wise and patch-wise approaches.

Furthermore, a transformer encoder with cross-layer fusion (CAF) was designed for spectral-spatial feature extraction. In addition, the group-wise spectral embedding (GSE) module was designed to remove local spectral profiles. Dual-view spectral and global spatial feature fusion (DSGSF) has two spectral and spatial feature extraction subnetworks. The first subnetwork is based on an encoder and decoder for global spatial feature extraction, while the second spectral subnetwork extracts spectral features. MF extracts high-dimensional spatial and spectral features using 2D and 3D convolution blocks. In addition, it performs morphological dilation operations in the transformer block to increase the interaction between the CLS and HSI tokens. In the SS1DSwin network consists of a Group feature tokenization module (GFTM) for token embedding and a 1DSwin Transformer block for global attention of the spectral-spatial features. The performance of the CNN-based models is relatively lower. 2D-CNN extracts only spatial features, while 3D-CNN extracts spectral features because this performance is superior in several land covers.

Moreover, HybridSN utilized the 2D-CNN and 3D-CNN for joint spatial and spectral features and demonstrated superior performance in several classes. Further, DSGSF improved performance through enhanced spatial features extracted from the encoder and decoder blocks and spectral features via the CNN module. However, traditional CNN-based models extract shallow high-dimensional features and ignore the edge features. ViT-based models have better quantitative results compared to traditional CNN models.

In CSIL, the inclusion of two transformer blocks has improved the classification accuracy. At the same time, computational costs have increased. The SF method utilized cross-layer fusion to improve the global co-relation of the features. However, costs increased due to additional learnable embedding. In the MF method, dilation-based spectral and spatial morphological operations are performed, which improved classification performance. The proposed method OA on the PU dataset is 3.1% better, and the kappa value is 2.1% better than SS1DSwin. Furthermore, OA is 1.39% better on the IP dataset than SS1DSwin.

Furthermore, the proposed model showed remarkable performance on the SA and Botswana datasets and obtained 97.52% and 89.45% OA, respectively. LSANet obtained the highest OA in the corn, grass-pasture, grass-pasture-mowed, hay-windrowed, soyabean-notill, soyabean-clean, woods and stone-steel-towers classes. Furthermore, on the Botswana dataset, the model respectively achieved 2.22% and 1.99% higher kappa and OA compared to SS1DSwin. Moreover, on the SV dataset, LSANet obtained 96.17% and 97.52% kappa and OA, respectively, which is better than other methods.

## 4.4. Visual results

The visual results of the proposed LSANet and 2D-CNN [32], 3D-CNN [33], HybridSN [34], CSIL [35], SpectralFormer(SF) [36], DSGSF [37], morphFormer (MF) [38] and SS1DSwin [39] on the PU, IP, SV and Botswana datasets are depicted in Figs 8–11. The region of interest (ROI) is selected and zoomed three times for better views of the Soyabean-clean and Stone-Steel-Towers of the IP datasets. Furthermore, the classification map of the Bare Soil and Bitumen class of the PU dataset is highlighted. In the classification map of the SV, Grapes_untrained class is highlighted and on the Botswana dataset floodplain grasses1 and firescar2 ROI is zoomed three times. The 2D-CNN and 3D-CNN classification map has large areas of noise and suffers from severe oversmoothing in the boundary region. The HybridSN and DSGSF improved the visual results in several classes, however suffer from smoothing issues. ViT based methods have clear classification maps but lack in differentiability of similar classes and showed overlapping. The proposed method showed classification maps close to GT except in a few classes.

## 4.5. Ablation studies

In this section, several hyperparameters that affect model performance on the IP, PU, SV and Botswana datasets are discussed in detail.

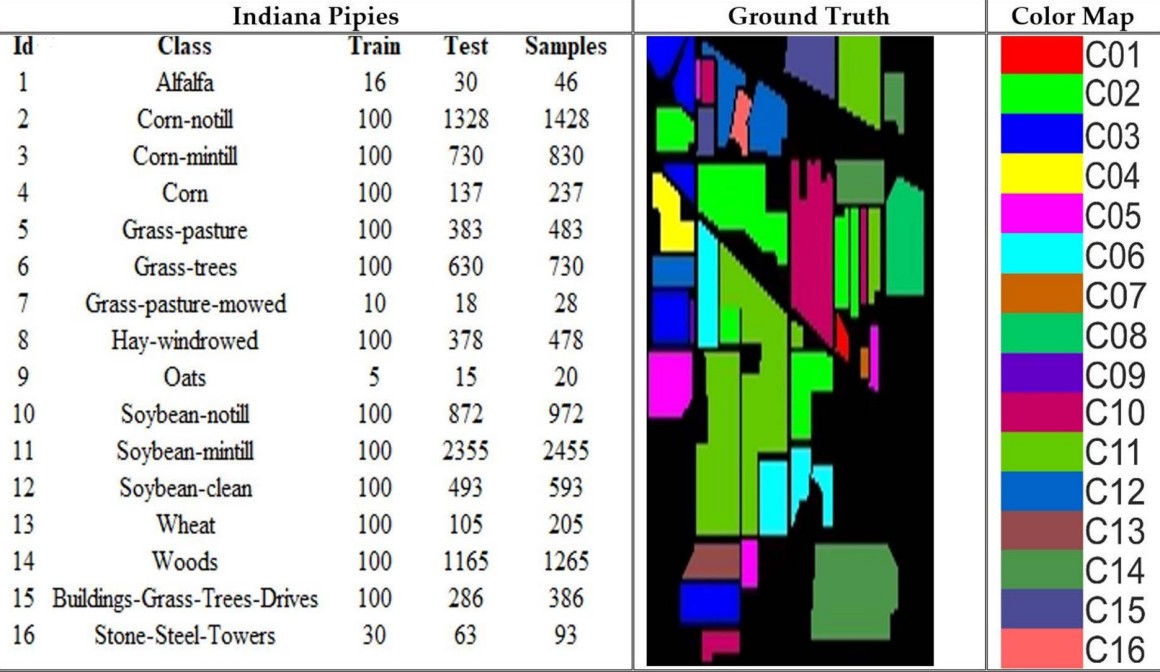

**Fig 5. Details of the Indiana Pines (IP) dataset with samples and color coding.**

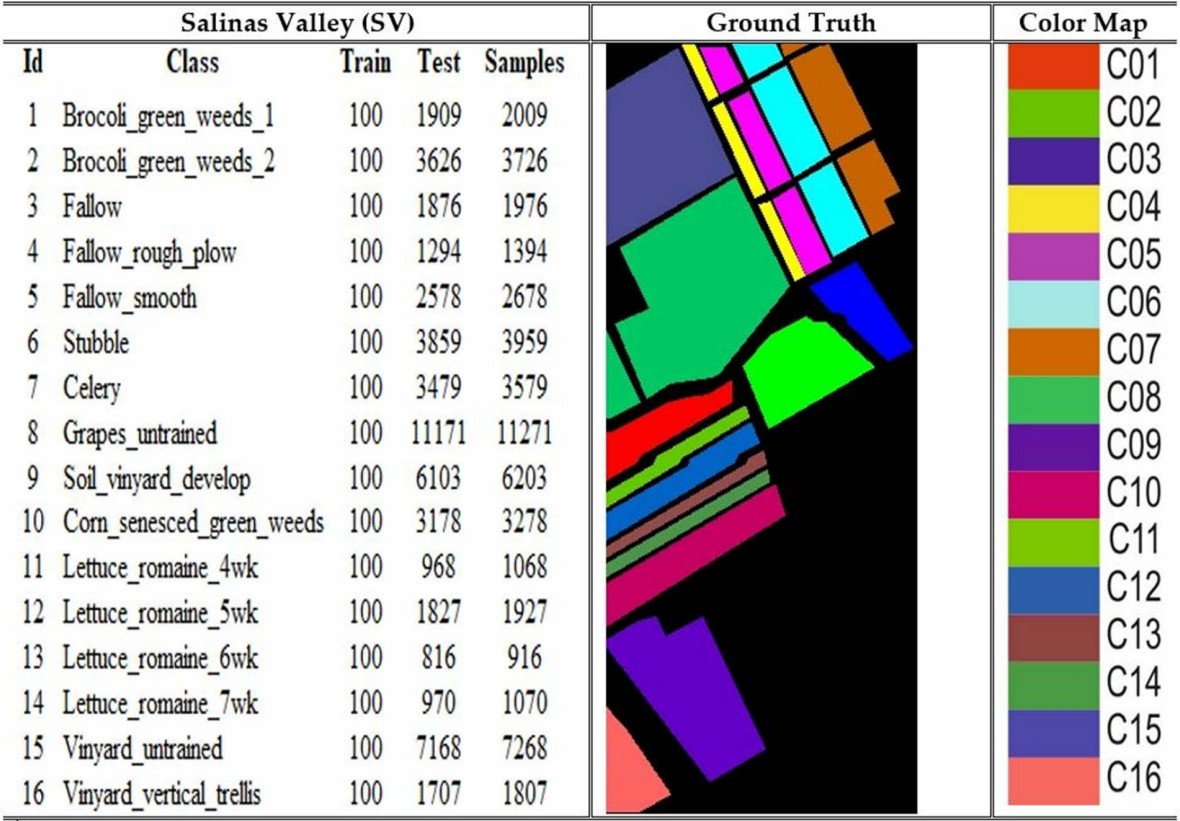

**Fig 6. Details of the Salinas Valley (SV) dataset with samples samples and color coding.**

**4.5.1. Effect of limpid size.** The size of the limpid greatly affects the contextual information and accuracy of the model. An experiment was conducted by varying the size in the range of 1–9 in the four stages of the transformer encoder. After the value of 9, the model did not gain significant improvement in land cover classification accuracy, as shown in Fig 12. However, the number of flops increases with the increase in limpid size, with the highest accuracy achieved for a size of 7 on the IP, PU, SV and Botswana datasets.

**4.5.2. Effect of different components.** The results of the ablation study using different components of the proposed LSANet on the four datasets are presented in Table 5. Table 5 shows that the OA accuracy of the ViT on the PU and SV datasets is 90.18% and 91.67%, respectively. On the other hand, 2D-CNN+ ViT improved the OA and Kappa values in the datasets. The 2D-CNN extracted high-dimensional local spatial features, and ViT provided global context. Furthermore, 3D-CNN+ViT achieved more than 2% improvement in the OA on PU, IP and SV datasets due to the capability of 3D-CNN towards spatial and spectral feature extraction. Moreover, 3D-CNN+2D-CNN+ViT achieved the highest OA accuracy and Kappa value in all the datasets due to improved spatial spectral features and global attention provided by the LS-attention based ViT encoder.

**4.5.3. Effect of batch size and learning rate.** The edge and boundary regions of the images require extensive parameter tuning for classification [40,41]. In addition, focusing on the edge and boundary regions can enhance object detection [42]. Hyperparameters, such as batch size and learning rate, are important choices for stable training and generalization of the model performance. Generally, a larger batch size leads to faster convergence of the model. At the same time, it might miss the fine-grained details of the hyperspectral data [43]. As we can notice in Fig 13(b), with

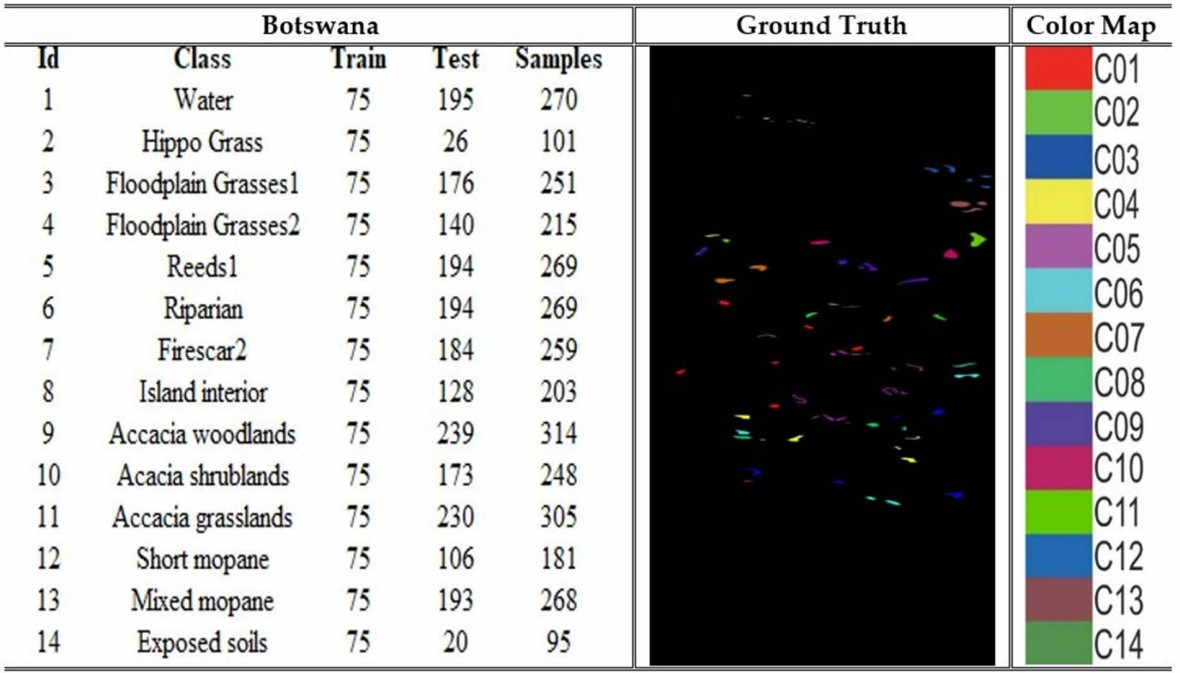

**Fig 7. Details of the Botswana dataset with samples samples and color coding.**

**Table 1. Performance comparison on the PU dataset.**

| CNN Based methods | | | | | ViT based methods | | | | |
|---|---|---|---|---|---|---|---|---|---|
| Cld. | 2D-CNN | 3D-CNN | HybridSN | DSGSF | CSIL | SF | MF | SS1DSwin | LSATNet |
| 1 | 72.10 | 74.21 | 75.51 | 84.15 | 90.79 | 95.29 | 96.95 | 93.25 | **97.55** |
| 2 | 67.24 | 67.16 | 72.65 | 88.35 | 93.75 | 92.61 | 95.53 | **98.73** | 98.28 |
| 3 | 58.78 | 72.47 | 78.35 | 63.61 | 83.03 | 82.62 | 87.77 | 93.43 | **94.47** |
| 4 | 70.30 | 76.29 | 82.23 | 90.37 | 90.15 | 93.27 | 90.13 | **96.57** | 95.96 |
| 5 | 81.20 | 84.32 | 86.19 | 92.53 | 94.95 | 93.61 | **98.57** | 92.13 | 96.79 |
| 6 | 82.19 | 84.62 | 87.53 | 81.97 | 96.39 | 90.43 | 94.87 | 96.07 | **96.91** |
| 7 | 75.38 | 82.98 | **94.21** | 82.43 | 85.57 | 87.81 | 85.55 | 91.54 | 93.25 |
| 8 | 65.37 | 72.68 | 76.85 | 77.67 | 88.29 | 91.59 | 95.79 | 97.71 | **99.86** |
| 9 | 72.35 | 78.69 | 74.85 | **97.13** | 91.51 | 82.53 | 93.87 | 90.63 | 96.46 |
| AA | 71.65 | 82.47 | 80.97 | 84.25 | 90.49 | 89.97 | 93.22 | 94.45 | **96.61** |
| OA | 73.18 | 83.52 | 82.19 | 85.57 | 92.37 | 90.29 | 95.71 | 95.86 | **98.78** |
| Kappa | 70.58 | 79.45 | 78.67 | 83.17 | 89.62 | 88.05 | 91.27 | 93.95 | **96.05** |

a batch size of 16, the proposed model accuracy on the PU, IP, SV, and Botswana datasets is lower. However, OA increases gradually with the batch size, and the highest value is obtained with a batch size of 64. Furthermore, OA slightly decreases with the further increase in batch size. The learning rate determines the number of iterations required for the model's stable training. Higher learning rates may lead to sensitivity to noise, and lower rates may increase the training iteration for stable performance. Fig 13(a) shows that the model achieved the best classification accuracy with a learning rate of $3e^{-4}$.

**Table 2. Quantitative results comparison on the IP dataset.**

| CNN Based methods | | | | | ViT based methods | | | | |
|---|---|---|---|---|---|---|---|---|---|
| CId. | 2D-CNN | 3D-CNN | HybridSN | DSGSF | CSIL | SF | MF | SS1DSwin | LSATNet |
| 1 | 72.87 | 68.26 | 65.18 | **97.86** | 77.85 | 78.71 | 84.51 | 96.52 | 97.23 |
| 2 | 65.31 | 67.69 | 76.65 | 80.49 | 81.26 | 83.73 | 90.37 | **97.36** | 96.01 |
| 3 | 54.39 | 62.71 | 78.35 | 72.71 | 64.87 | 77.79 | **98.78** | 97.41 | 97.88 |
| 4 | 70.15 | 72.96 | 74.18 | 79.73 | 82.75 | 80.52 | 91.85 | 96.28 | **97.52** |
| 5 | 62.72 | 74.63 | 65.19 | 72.24 | 77.32 | 79.63 | 91.36 | 96.12 | **98.26** |
| 6 | 75.60 | 61.23 | 79.18 | 78.79 | 80.23 | 85.57 | 88.57 | **97.23** | 96.81 |
| 7 | 59.16 | 57.96 | 67.10 | 73.21 | 73.49 | 80.32 | 92.23 | 99.15 | **98.40** |
| 8 | 78.61 | 75.37 | 82.15 | 88.75 | 92.27 | 94.26 | 89.76 | 96.63 | **99.25** |
| 9 | 56.28 | 64.85 | **93.87** | 70.63 | 76.83 | 80.68 | 83.89 | 87.72 | 91.31 |
| 10 | 62.67 | 70.18 | 72.36 | 71.08 | 71.40 | 82.72 | 84.91 | 96.47 | **99.61** |
| 11 | 76.17 | 74.96 | 77.19 | 81.36 | 83.87 | 85.41 | 95.68 | **98.75** | 96.41 |
| 12 | 62.76 | 68.19 | 72.19 | 76.81 | 73.29 | 83.98 | 92.86 | 94.52 | **97.72** |
| 13 | 69.30 | **98.14** | 76.38 | 74.15 | 79.85 | 79.47 | 92.98 | 98.23 | 97.43 |
| 14 | 72.21 | 79.82 | 78.60 | 81.94 | 84.91 | 92.40 | 94.91 | 96.82 | **99.50** |
| 15 | 58.96 | 72.17 | 70.53 | 73.81 | 74.69 | 82.76 | 93.45 | 95.98 | **98.98** |
| 16 | **98.38** | 68.92 | 76.87 | 77.67 | 81.81 | 83.94 | 91.91 | 94.72 | 97.82 |
| AA | 68.47 | 71.13 | 75.37 | 78.20 | 78.54 | 83.24 | 91.12 | 96.24 | **97.50** |
| OA | 70.10 | 72.65 | 76.59 | 79.41 | 80.31 | 85.62 | 93.42 | 97.28 | **98.67** |
| Kappa | 67.17 | 68.21 | 74.69 | 76.32 | 76.28 | 81.86 | 90.62 | 95.89 | **97.10** |

**Table 3. Performance comparison on the Botswana dataset.**

| CNN Based methods | | | | | ViT based methods | | | | |
|---|---|---|---|---|---|---|---|---|---|
| CId. | 2D-CNN | 3D-CNN | HybridSN | DSGSF | CSIL | SF | MF | SS1DSwin | LSATNet |
| 1 | 55.24 | 52.17 | 62.78 | 70.12 | 68.75 | 72.34 | 65.19 | 78.56 | **82.34** |
| 2 | 46.78 | 50.39 | 58.96 | 65.73 | 67.18 | 69.72 | 64.13 | **78.84** | 78.60 |
| 3 | 72.10 | 76.57 | 82.10 | **80.54** | 79.51 | 77.19 | 78.32 | 79.26 | 80.14 |
| 4 | 65.28 | 84.12 | 81.49 | 82.76 | 87.92 | 84.15 | 86.87 | 87.64 | **90.18** |
| 5 | 76.38 | 78.16 | 80.36 | 79.17 | **92.11** | 86.74 | 90.25 | 91.87 | 91.89 |
| 6 | 84.31 | 82.18 | **85.48** | 83.93 | 85.98 | 82.79 | 82.74 | 84.13 | 85.17 |
| 7 | 78.52 | 76.38 | 74.96 | 81.83 | 78.67 | **88.71** | 80.31 | 85.27 | 87.84 |
| 8 | 82.85 | **88.12** | 87.19 | 85.89 | 86.21 | 85.79 | 84.65 | 87.78 | 86.95 |
| 9 | 84.37 | 90.87 | 86.82 | 87.98 | 93.45 | 91.27 | **94.87** | 92.76 | 93.18 |
| 10 | 75.18 | 81.26 | 84.71 | 75.93 | 84.83 | 88.67 | 86.27 | 90.28 | **94.53** |
| 11 | **92.13** | 84.87 | 86.63 | 84.79 | 88.53 | 90.14 | 86.49 | 87.84 | 91.76 |
| 12 | 66.58 | 75.62 | 78.14 | 82.35 | 79.18 | 84.87 | 78.24 | 83.42 | **85.69** |
| 13 | 80.42 | 82.67 | 84.29 | 79.89 | 85.46 | 87.97 | 84.37 | 92.94 | 91.87 |
| 14 | 68.45 | 78.85 | 76.72 | 75.79 | 72.98 | 83.27 | 76.89 | 80.25 | **87.38** |
| AA | 73.47 | 77.03 | 76.95 | 79.76 | 82.20 | 83.83 | 81.40 | 85.77 | **87.68** |
| OA | 75.71 | 79.28 | 78.86 | 81.12 | 83.98 | 85.67 | 82.74 | 87.23 | **89.45** |
| Kappa | 72.13 | 76.52 | 75.17 | 78.37 | 81.59 | 82.19 | 80.78 | 84.15 | **86.14** |

**Table 4. Performance comparison on the SV dataset.**

| | CNN Based methods | | | | ViT based methods | | | | |
|---|---|---|---|---|---|---|---|---|---|
| Cld. | 2D-CNN | 3D-CNN | HybridSN | DSGSF | CSIL | SF | MF | SS1DSwin | LSATNet |
| 1 | 75.47 | 73.65 | 72.51 | 77.10 | 86.25 | 85.52 | 85.23 | **96.63** | 96.58 |
| 2 | 65.25 | 68.37 | 73.19 | 72.15 | 79.36 | 87.67 | 84.84 | 92.24 | **97.51** |
| 3 | 73.35 | 76.69 | 78.60 | 82.63 | 86.18 | 92.42 | **97.73** | 90.52 | 97.42 |
| 4 | 76.28 | 81.37 | **98.89** | 84.74 | 86.72 | 81.82 | 97.65 | 87.21 | 97.25 |
| 5 | 74.23 | 80.39 | 84.75 | **97.96** | 86.87 | 89.76 | 96.26 | 96.87 | 96.12 |
| 6 | 81.12 | **97.89** | 82.37 | 89.18 | 89.64 | 86.98 | 97.64 | 95.29 | 96.45 |
| 7 | 68.27 | 82.14 | 76.19 | 85.52 | 87.53 | 82.84 | 98.28 | **99.46** | 96.21 |
| 8 | 82.74 | 79.86 | 85.18 | 84.73 | 86.82 | 87.87 | 94.52 | 91.63 | **99.76** |
| 9 | 73.42 | 80.51 | 85.69 | 86.78 | 87.29 | **90.69** | 89.82 | 85.25 | 89.52 |
| 10 | 82.78 | 75.92 | 76.84 | 77.49 | 79.48 | 85.34 | 86.85 | 91.83 | **95.25** |
| 11 | 56.82 | 83.61 | 87.94 | 88.53 | **97.56** | 89.57 | 97.47 | 95.23 | 95.51 |
| 12 | 83.89 | 84.72 | 86.45 | 84.98 | 92.25 | 85.83 | 96.63 | 96.21 | **99.25** |
| 13 | 87.28 | 78.92 | 85.73 | 87.86 | 86.24 | 96.75 | 97.23 | 96.45 | 97.71 |
| 14 | 86.79 | 84.19 | 88.96 | 92.17 | 86.75 | 93.23 | 94.67 | 95.78 | **99.68** |
| 15 | **96.17** | 87.83 | 83.54 | 94.18 | 94.46 | 86.56 | 93.62 | 89.24 | 95.42 |
| 16 | 78.83 | 92.87 | 90.64 | 88.97 | 86.85 | 93.63 | 96.28 | 92.78 | **98.49** |
| AA | 77.67 | 81.80 | 83.60 | 85.93 | 87.51 | 88.22 | 94.04 | 93.28 | **96.75** |
| OA | 78.96 | 82.57 | 85.19 | 87.14 | 88.32 | 90.51 | 95.63 | 94.45 | **97.52** |
| Kappa | 76.49 | 80.65 | 82.17 | 83.89 | 86.14 | 87.37 | 93.51 | 92.91 | **96.17** |

**4.5.4. Effect of CPE and zero padding.** CPE and zero padding in the LSANet are important factors for improving the classification performance. We summarize model performance on IP, PU, SV and Botswana datasets with several combinations of these parameters, as shown in Table 6. Table 6 shows that without CPE and zero padding, classification accuracy on the IP dataset is 95.19%, whereas CPE improved performance by more than 1.5%. Furthermore, the model achieved 98.67% accuracy with CPE and zero padding on the IP dataset. Moreover, on the PU dataset, the model gained a 0.16% performance improvement with CPE and zero padding. At the same time, on the SV dataset, the model achieved an OA of 95.26% with CPE and 96.29% with zero padding, respectively. LSANet achieved the highest performance on all datasets using both CPE and zero padding.

**4.5.5. Training time analysis.** The trainable parameters and samples presented in the land covers influence the models training time. In the proposed study, the training times of each model under the same experimental conditions were calculated on the PU, IP, SV and Botswana datasets. As shown in Fig 14, it can observed that 2D-CNN takes less time on all the datasets due to less trainable parameters. Meanwhile, 3D-CNN has a high training time due to large numbers of training parameters. Furthermore, Hybrid CNN, which has 3D and 2D convolutional layers, exhibits higher train times than 2D-CNN. ViT-based methods, SF and MF, have higher computation times than classical CNN, except for 3D-CNN, due to the computation of attention in the encoder. The SS1DSwin has relatively lower training times due to the attention calculation through 1D-CNN layers. The proposed LSANet is designed using lightweight 3D-CNN and 2D-CNN, and attention is calculated through parallel row- and column-wise. Due to this, training times are smaller than the majority of methods.

**4.5.6. Performance comparison with different attention.** The experiments are conducted using self-attention (SA), axial self-attention (ASA), and cross self-attention (CSA) by replacing the LSA in the proposed model on the PU, IP, SV and Botswana datasets. The experimental settings were kept the same, as discussed in section 4.2. After training, OA and Kappa values on each dataset are calculated, as shown in Table 7. The ViT + SA achieved 91.56% and 93.19% Kappa

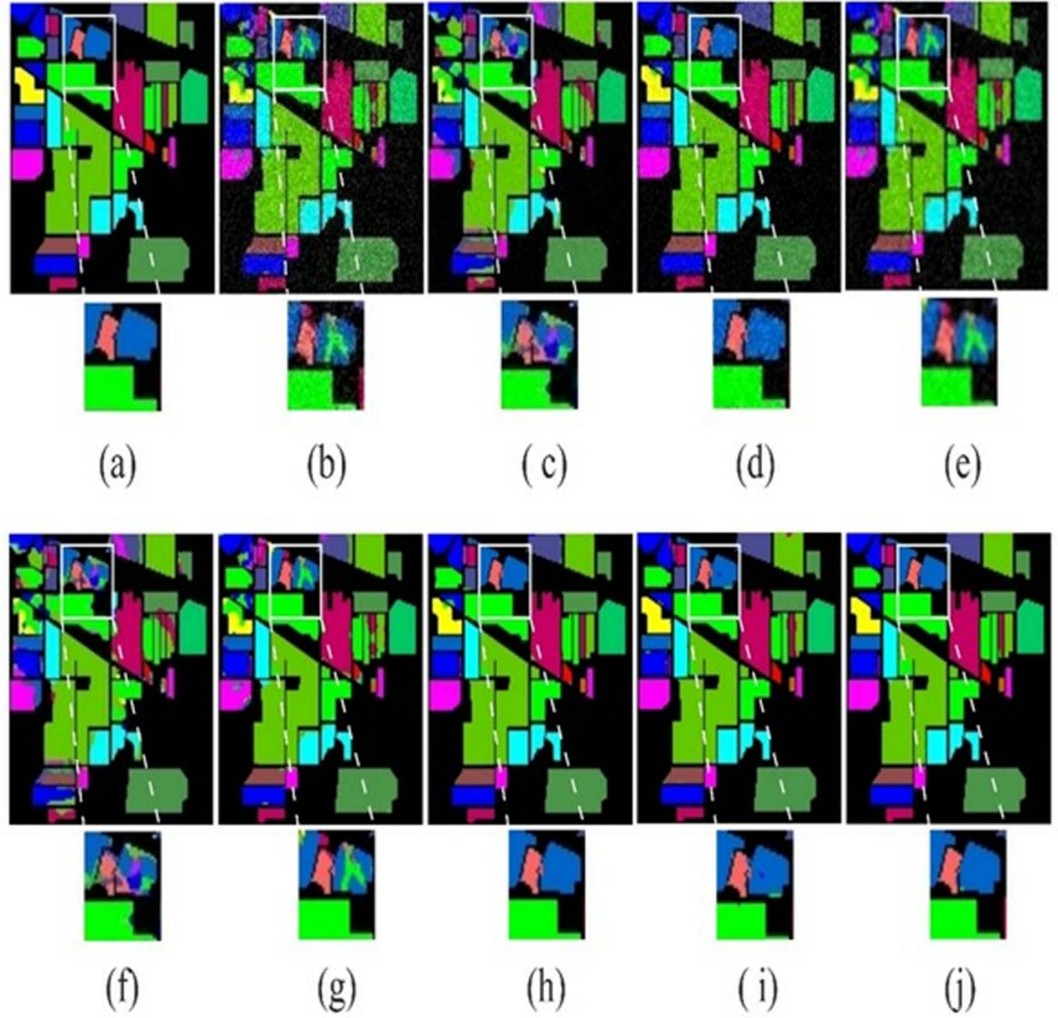

**Fig 8. The visual map on PU dataset.** (a)GT (b)2D-CNN (c)3D-CNN (d)HybridSN (e)CSIL (f)SF (g) DSGSF (h)MF (i) SS1DSwin and (j) LSANet.

values on the PU and IP datasets, respectively. Meanwhile, the ViT+ASA respectively obtained Kappa values of 92.07% and 94.16% in the PU and SV datasets. The ViT+CSA improved the Kappa values by 2.75%, 2.14% and 0.88% on the PU, Botswana and SV datasets. Moreover, our ViT+LSA obtained Kappa values of 96.05%, 97.10% and 86.14% on the PU, IP and Botswana datasets. Furthermore, the computational complexity of the SA is $O\left(h^2 \times w^2 \times d_k\right)$, which grows quadratic with the size of the feature map. Whereas, in the ASA, attention is calculated using row- and column-wise sequentially and complexity is reduced to $O\left(h^2 \times w \times d_k + h \times w^2 \times d_k\right)$. Meanwhile, in the CSA attention is calculated between two feature maps, which is $O\left(h_1 \times w_1 \times h_2 \times w_2 \times d_k\right)$. Moreover, our LSA calculates parallel attention in the row and column of each block and has complexity, $O\left(\frac{h}{u_r} \times \frac{w}{u_c} x\left(u_r \times w + u_c \times h\right)^2 \times d_k\right)$, where, h,w is the input shape, $d_k$ is the dimension of the query/key, and $u_r, u_c$ is the size of the limpid. The SA mechanism in the ViT calculates the attention score across all token pairs and can provide global attention. However, it may miss the local mid-level features essential for HSI classification. Moreover, ASA calculates attention in rows and columns at a time. Due to this, the diagonal features may be missing. At the same time, CSA has a cross pattern that is broader along the axes but narrower in terms of directional coverage. Lastly, LSA calculates attention in all directions and is more adaptive to providing contextual information.

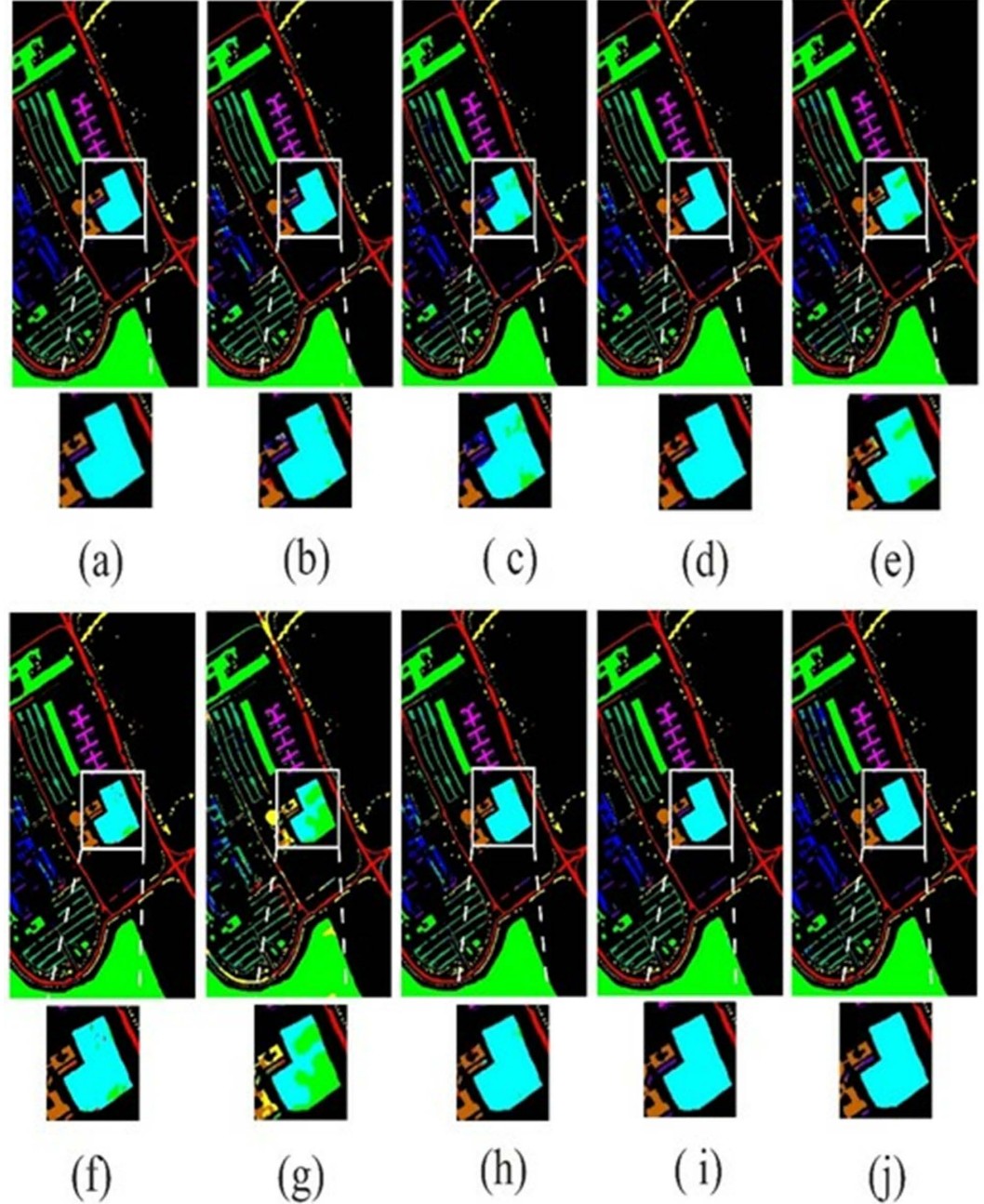

**Fig 9. Illustration of the visual map on IP dataset.** (a)GT (b)2D-CNN (c)3D-CNN (d)HybridSN (e)CSIL (f)SF (g) DSGSF (h)MF (i) SS1DSwin and (j) LSANet.

**4.5.7. Performance evaluation using different encoding techniques.** An experiment is conducted using FPE and LPE on the PU, IP, SV and Botswana datasets, and the results are presented in Table 8. Table 8 shows that the FPE using the sine function has an OA of 94.86% and a Kappa value of 93.18%. At the same time, it achieved 86.15% and 93.28% OA on the SV and Botswana datasets, respectively. The LPE achieved better OA and Kappa values than FPE on

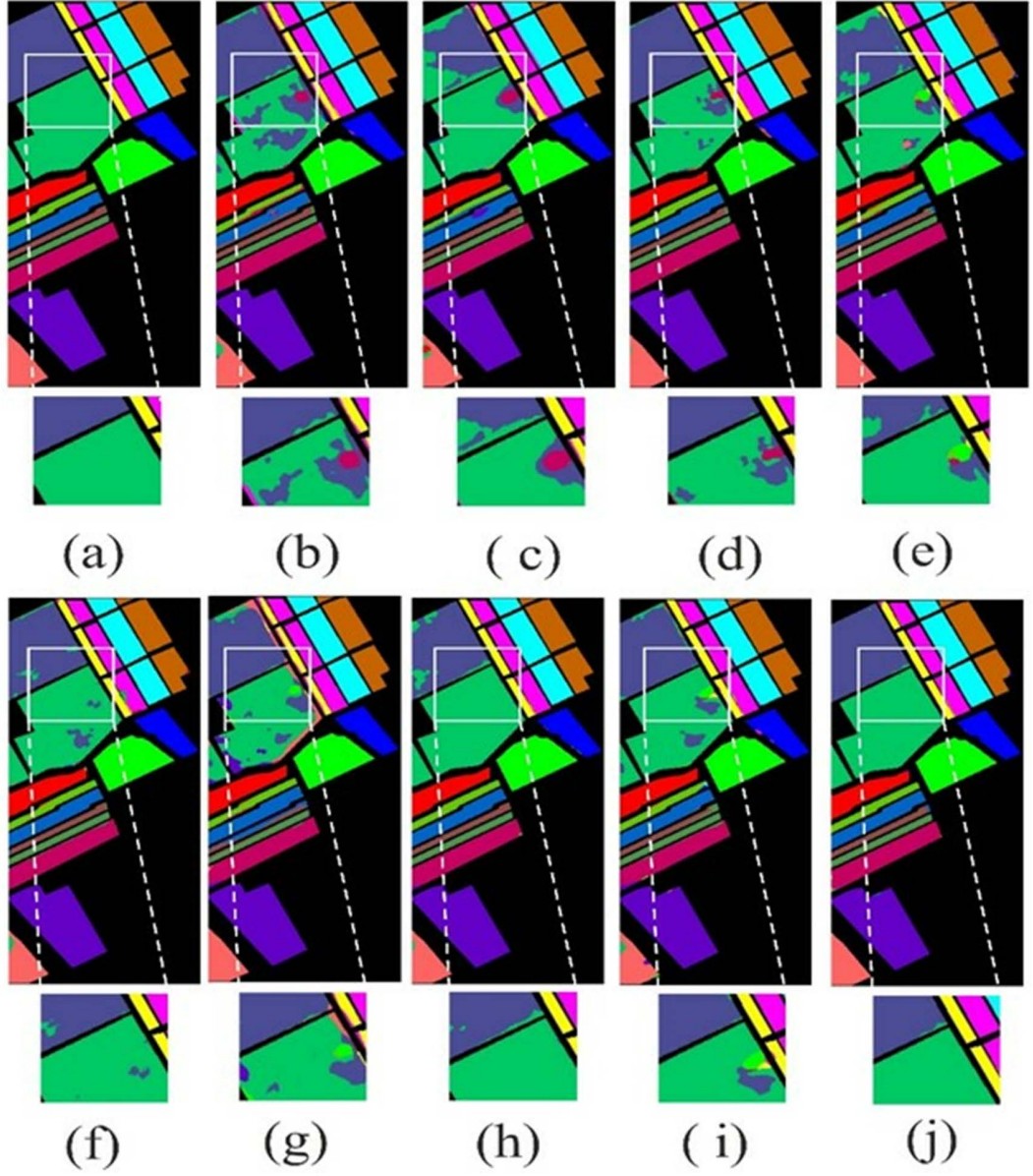

**Fig 10. The visual map on SV dataset.** (a)GT (b)2D-CNN (c)3D-CNN (d)HybridSN (e)CSIL (f)SF (g) DSGSF (h)MF (i) SS1DSwin and (j) LSANet.

all the datasets. Moreover, the CPE obtained OA of 98.78% and 89.45% on the PU and SV datasets, respectively. Fixed positional encoding (FPE) , e.g., sinusoidal, uses a sine or cosine function to generate the image encodings at different frequencies. Another fixed positional encoding technique, learned positional encoding (LPE), provides a model to learn positional encodings from the positional encoding vectors during training. However, after training, it remains fixed. Our CPE generates dynamic positional encodings using depth-wise convolution from the input feature map, which makes them efficient and flexible across different resolutions.

**4.5.8. Parameters and flops comparison.** The trainable parameters in millions (M) and flops in Gigaflops (G) of each model based on 30 bands of the datasets are depicted in Table 9. Table 9 shows that the 3D-CNN model has the highest

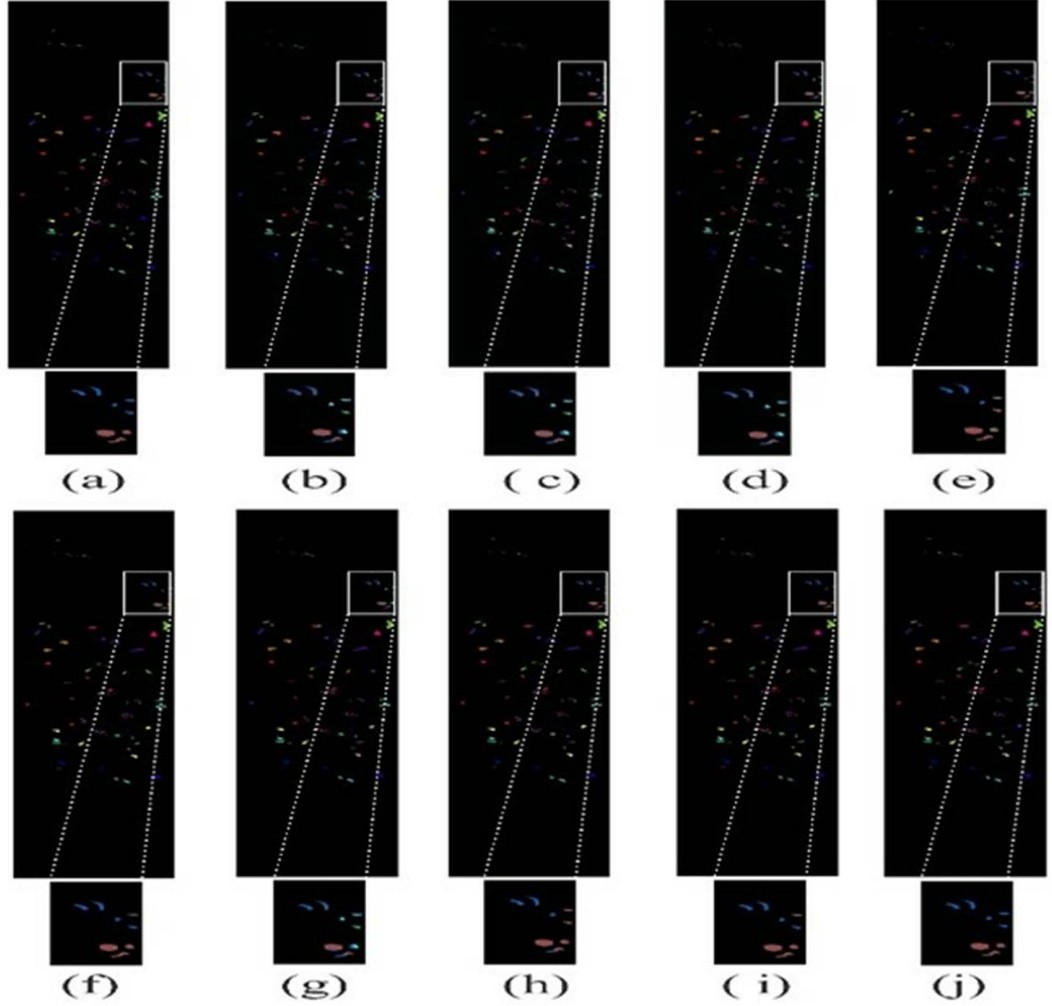

**Fig 11. The visual map on Botswana dataset.** (a)GT (b)2D-CNN (c)3D-CNN (d)HybridSN (e)CSIL (f)SF (g) DSGSF (h)MF (i) SS1DSwin and (j) LSANet.

number of trainable parameters and the 2D-CNN has the lowest. At the same time, both 2D-CNN and 3D-CNN have high flop values. The transformer-based method,s CSIL, SpectralFormer, DSGSF, MorphFormer and SS1DSwin, also have high flop values. For instance, SS1DSwin has 26.05 M trainable parameters. The proposed LSANet has 2.3M trainable parameters and 1.27 G flops.

## 5. Conclusion

Hyperspectral data is characterized by the unique challenge of limited spectral bands that provide extensive spectral information but reduced spatial detail. CNNs have made incredible progress in collecting high-level contextual information for remote sensing applications, particularly hyperspectral image classification tasks. However, CNNs lack in adequately characterizing sequence features of spectral signatures and capturing the global characteristics of complete HSIs with local kernels. In this study, LSANet is proposed, which overcomes these constraints by combining global attention mechanisms for spatial and spectral information generated by CNNs. A conditional position encoding (CPE) is utilized that

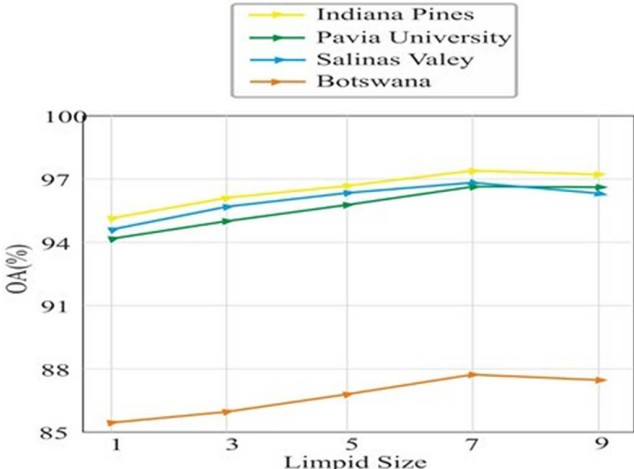

**Fig 12. Illustration of limpid size on IP, PU, SV and Botswana datasets.**

**Table 5. Performance evaluation using different components.**

| Model | PU OA Kappa | IP OA Kappa | Botswana OA Kappa | SV OA Kappa |
|---|---|---|---|---|
| ViT | 90.18 89.47 | 91.25 90.73 | 83.46 82.09 | 91.67 90.85 |
| 2D-CNN+ViT | 92.60 91.72 | 92.30 91.05 | 85.27 83.14 | 93.25 92.32 |
| 3D-CNN+ViT | 95.12 93.87 | 94.75 93.20 | 86.53 85.10 | 95.36 94.19 |
| 3D-CNN+2D-CNN+ViT | 98.78 96.05 | 98.67 97.10 | 89.45 86.14 | 97.52 96.17 |

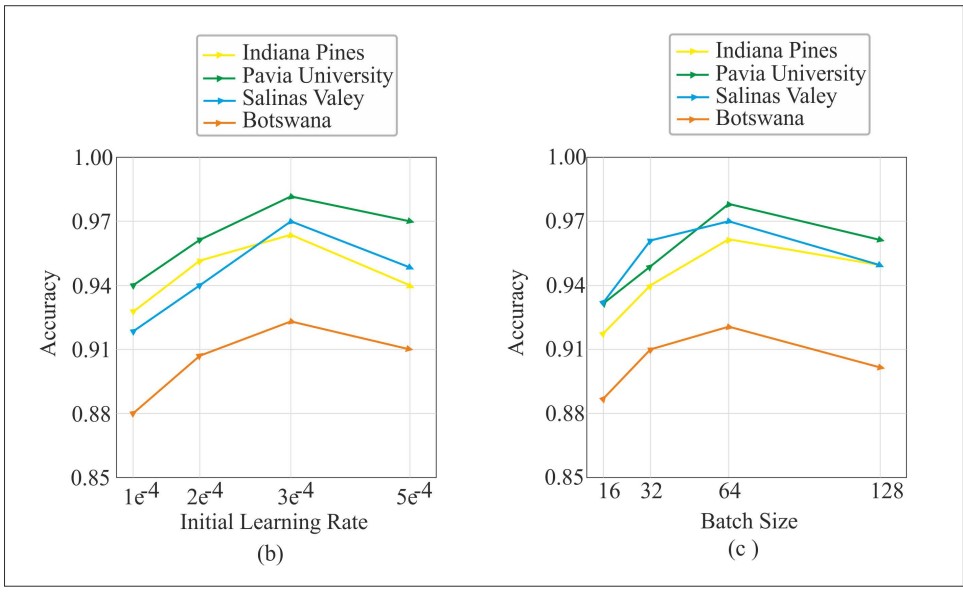

**Fig 13. Illustration of the (a) learning rate and (b) batch size on the IP, PU, SV and Botswana datasets.**

**Table 6. LSANet performance on IP dataset with CPE and zero padding.**

| CPE | Padding | IP OA | PU OA | SV OA | Botswana OA |
|---|---|---|---|---|---|
| ✓ | x | 96.34% | 97.02% | 95.26% | 85.64% |
| x | ✓ | 97.01% | 97.18% | 96.29% | 87.30% |
| x | x | 95.19% | 96.06% | 95.98% | 86.76% |
| ✓ | ✓ | 98.67% | 98.78% | 97.52% | 89.45% |

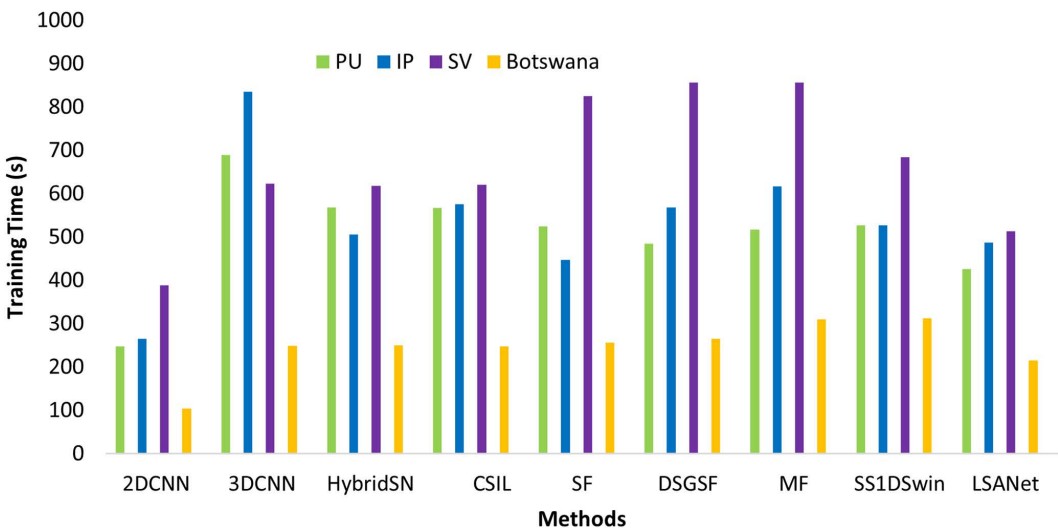

**Fig 14. Training time comparison on PU, IP, SV, and Botswana datasets.**

**Table 7. Performance comparison using different attention mechanisms.**

| Model | PU OA Kappa | IP OA Kappa | Botswana OA Kappa | SV OA Kappa |
|---|---|---|---|---|
| ViT+SA | 93.29 91.56 | 94.40 93.19 | 85.38 81.95 | 92.86 91.45 |
| ViT+ASA | 94.95 92.07 | 95.26 93.49 | 85.98 82.64 | 95.37 94.16 |
| ViT+CSA | 96.09 94.82 | 96.25 95.18 | 87.27 84.78 | 96.34 95.04 |
| ViT+LSA | 98.78 96.05 | 98.67 97.10 | 89.45 86.14 | 97.52 96.17 |

**Table 8. Performance comparison using different encoding techniques.**

| Encoding | PU OA Kappa | IP OA Kappa | SV OA Kappa | Botswana OA Kappa |
|---|---|---|---|---|
| FPE (sine) | 94.86 93.18 | 95.24 94.09 | 86.15 83.50 | 93.28 92.07 |
| LPE | 95.27 94.08 | 96.82 94.97 | 87.64 85.16 | 95.03 94.29 |
| CPE | 98.78 96.05 | 98.67 97.10 | 89.45 86.14 | 97.52 96.17 |

dynamically creates tokens, which improves the ability of the model to identify subtle patterns in data. Furthermore, by separating feature maps into smaller regions for token interaction, our LS-attention strategy provides a more comprehensive contextual representation than earlier attention mechanisms. The superiority of the model is tested on four datasets and achieves OAs of 98.78%, 98.67%, 97.52% and 89.45% on PU, IP, SV and Botswana datasets, respectively. One

**Table 9. Parameters and flops comparison.**

| Model | Parameters (M) | Flops (G) |
|---|---|---|
| 2D-CNN | 0.40 | 2.69 |
| 3D-CNN | 71.17 | 2.57 |
| HybridSN | 5.12 | 0.26 |
| CSIL | 0.70 | 1.84 |
| SpectralFormer | 0.72 | 1.87 |
| DSGSF | 0.82 | 1.01 |
| MorphFormer | 0.64 | 1.29 |
| SS1DSwin | 26.05 | 0.69 |
| Proposed LSANet | 2.34 | 1.27 |

major limitation includes computational costs of the attention mechanism that can be reduced through other approaches. Another constraint is zero padding and CPE to improve the classification performance. In the future, transformer-based architectures with advanced techniques, e.g., efficient attention and self-supervised learning, can be designed to better fit the HS data classification task. Additionally, a lightweight ViT network can be designed to decrease complexity, while maintaining performance. Furthermore, more physical features of the spectral bands based on past information can be extracted to improve the classification performance.

## Author contributions

**Conceptualization:** Dhirendra Prasad Yadav, Bhisham Sharma.

**Data curation:** Dhirendra Prasad Yadav.

**Formal analysis:** Bhisham Sharma.

**Investigation:** Deepak Kumar, Anand Singh Jalal, Bhisham Sharma.

**Methodology:** Dhirendra Prasad Yadav.

**Project administration:** Deepak Kumar, Anand Singh Jalal, Panos Liatsis.

**Resources:** Panos Liatsis.

**Software:** Panos Liatsis.

**Supervision:** Anand Singh Jalal, Bhisham Sharma.

**Validation:** Panos Liatsis.

**Writing – original draft:** Dhirendra Prasad Yadav, Deepak Kumar.

**Writing – review & editing:** Anand Singh Jalal, Bhisham Sharma, Panos Liatsis.

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
