## [Decision Letter · Decision Letter 0]

PONE-D-25-23717Leveraging Potential of Limpid Attention Transformer with Dynamic Tokenization for Hyperspectral Image ClassificationPLOS ONE

Dear Dr. Sharma,

Thank you for submitting your manuscript to PLOS ONE. After careful consideration, we feel that it has merit but does not fully meet PLOS ONE’s publication criteria as it currently stands. Therefore, we invite you to submit a revised version of the manuscript that addresses the points raised during the review process.

We look forward to receiving your revised manuscript.

Kind regards,

Bardia Yousefi, Ph.D.

Academic Editor

PLOS ONE

3. Thank you for uploading your study's underlying data set. Unfortunately, the repository you have noted in your Data Availability statement does not qualify as an acceptable data repository according to PLOS's standards.

Additional Editor Comments:

There are concerns about that could be addressed in a major revision

Reviewers' comments:

Reviewer's Responses to Questions

**Comments to the Author**

1. Is the manuscript technically sound, and do the data support the conclusions?

Reviewer #1: Yes

Reviewer #2: Yes

2. Has the statistical analysis been performed appropriately and rigorously? 

Reviewer #1: Yes

Reviewer #2: Yes

3. Have the authors made all data underlying the findings in their manuscript fully available?

Reviewer #1: Yes

Reviewer #2: Yes

4. Is the manuscript presented in an intelligible fashion and written in standard English?

Reviewer #1: Yes

Reviewer #2: Yes

5. Review Comments to the Author

Reviewer #1: Clarify the meaning of “limpid” in the context of attention—it’s not a standard term in machine learning, so defining it early would help.

Add a summary table or bullet points comparing the number of parameters and training time across models.

Consider explaining the intuition behind some design choices (e.g., choice of 3D-CNN size, limpid block configuration).

Reviewer #2: The reviewer appreciates authors for their research work. The paper meets the quality level of PLOS one and can be accepted for publication after addressing following major revisions. Thank you.

1. Convolutional neural networks (CNNs), limpid size attention network (LSANet) and other abbreviations should be written like this throughout the paper.

2. Also, usage of personal pronoun “We” should also be avoided throughout the paper.

3. Authors are suggested to solidify their introduction section further some latest work on the proposed topic are: “A deep recurrent learning-based region-focused feature detection for enhanced target detection in multi-object media”, “Enhancing image categorization with the quantized object recognition model in surveillance systems” and “Improved camouflaged detection in the large-scale images and videos with minimum boundary contrast in detection technique”.

4. Authors should include a clear problem statement and articulate how LSANet directly addresses an unmet challenge in HSI classification.

5. Improve resolution and labeling of all figures.

6. Add 2–3 sentences after each ablation result table to interpret key findings. Explain why each component contributes to performance improvement.

7. Justify why zero-padding was chosen over alternative techniques (e.g., reflection padding or learned embeddings) and its impact on edge classification.

6. PLOS authors have the option to publish the peer review history of their article (what does this mean? ). If published, this will include your full peer review and any attached files.

**Do you want your identity to be public for this peer review?** For information about this choice, including consent withdrawal, please see our Privacy Policy .

Reviewer #1: **Yes: ** Ghassan Abdul-Majeed

Reviewer #2: No

---

## [Author Response · Author response to Decision Letter 1]

23 Jun 2025

Please find the attached file for reviewers response.

---

## [Decision Letter · Decision Letter 1]

Leveraging Potential of Limpid Attention Transformer with Dynamic Tokenization for Hyperspectral Image Classification

PONE-D-25-23717R1

Dear Dr. Liatsis,

We’re pleased to inform you that your manuscript has been judged scientifically suitable for publication and will be formally accepted for publication once it meets all outstanding technical requirements.

Kind regards,

Bardia Yousefi, Ph.D.

Academic Editor

PLOS ONE

Additional Editor Comments (optional):

Reviewers' comments:

Reviewer's Responses to Questions

**Comments to the Author**

1. If the authors have adequately addressed your comments raised in a previous round of review and you feel that this manuscript is now acceptable for publication, you may indicate that here to bypass the “Comments to the Author” section, enter your conflict of interest statement in the “Confidential to Editor” section, and submit your "Accept" recommendation.

Reviewer #1: All comments have been addressed

Reviewer #2: All comments have been addressed

2. Is the manuscript technically sound, and do the data support the conclusions?

Reviewer #1: Yes

Reviewer #2: Yes

3. Has the statistical analysis been performed appropriately and rigorously? 

Reviewer #1: Yes

Reviewer #2: Yes

4. Have the authors made all data underlying the findings in their manuscript fully available?

Reviewer #1: Yes

Reviewer #2: Yes

5. Is the manuscript presented in an intelligible fashion and written in standard English?

Reviewer #1: Yes

Reviewer #2: Yes

6. Review Comments to the Author

Reviewer #1: The revised manuscript is much better and deserves acceptance for publication. The authors made significant efforts to address the comments of the reviewers.

Reviewer #2: Dear Reviewers, the reviewer is very much pleased to recommend the acceptance of revised manuscript in current form. Congratulations.

7. PLOS authors have the option to publish the peer review history of their article (what does this mean? ). If published, this will include your full peer review and any attached files.

**Do you want your identity to be public for this peer review?** For information about this choice, including consent withdrawal, please see our Privacy Policy .

Reviewer #1: **Yes: ** Ghassan Abdul-Majeed

Reviewer #2: No

---

## [Editor Report · Acceptance letter]

PONE-D-25-23717R1

PLOS ONE

Dear Dr. Liatsis,

I'm pleased to inform you that your manuscript has been deemed suitable for publication in PLOS ONE. Congratulations! Your manuscript is now being handed over to our production team.

Kind regards,

on behalf of

Dr. Bardia Yousefi

Academic Editor

PLOS ONE